# Swept coded aperture real-time femtophotography

Jingdan Liu [1,4,5], Miguel Marquez[1,5], Yingming Lai[1], Heide Ibrahim[1], Katherine Légaré[1], Philippe Lassonde[1], Xianglei Liu[1], Michel Hehn [2], Stéphane Mangin [2], Grégory Malinowski [2], Zhengyan Li[3], François Légaré[1] & Jinyang Liang [1] ✉

Single-shot real-time femtophotography is indispensable for imaging ultrafast dynamics during their times of occurrence. Despite their advantages over conventional multi-shot approaches, existing techniques confront restricted imaging speed or degraded data quality by the deployed optoelectronic devices and face challenges in the application scope and acquisition accuracy. They are also hindered by the limitations in the acquirable information imposed by the sensing models. Here, we overcome these challenges by developing swept coded aperture real-time femtophotography (SCARF). This computational imaging modality enables all-optical ultrafast sweeping of a static coded aperture during the recording of an ultrafast event, bringing full-sequence encoding of up to 156.3 THz to every pixel on a CCD camera. We demonstrate SCARF's single-shot ultrafast imaging ability at tunable frame rates and spatial scales in both reflection and transmission modes. Using SCARF, we image ultrafast absorption in a semiconductor and ultrafast demagnetization of a metal alloy.

Since Abramson's pioneering light-in-flight recording by holography[1], imaging ultrafast events in real time (i.e., in the time duration of the event's occurrence) has contributed to numerous studies in diverse scientific fields, including nuclear fusion[2], photon transport in scattering media[3], and radiative decay of molecules[4]. Because many of these ultrafast phenomena have timespans from femtoseconds to picoseconds, femtophotography—recording two-dimensional (2D) spatial information at trillions of frames per second (Tfps)—is indispensable for clearly resolving their spatiotemporal details[5]. Currently, femtophotography is mostly realized by using multi-shot approaches[6]. In data acquisition, each measurement captures a temporal slice by time gating[7,8], a spatiotemporal slice using ultrafast devices[9,10], or a certain number of time-stamped events using photon-counting cameras[11,12]. Then, repetitive measurements (with certain auxiliaries,

such as temporal or spatial scanning) are performed to construct a movie. However, these methods require the dynamic events under observation to be precisely reproducible, which renders them incapable of studying non-repeatable or difficult-to-reproduce ultrafast phenomena, such as femtosecond laser ablation[13], shock-wave interaction with living cells[14], and optical chaos[15].

To surmount these limitations, many single-shot ultrafast imaging techniques have been developed for direct observation of dynamic events in real time. Existing techniques can be generally grouped into the categories of passive detection and active illumination[16]. The former is propelled by disruptive hardware designs, such as an in-situ storage CCD[17], a shutter-stacked CMOS sensor[18], and a framing camera[19]. Nonetheless, thus far, these ultrafast sensors have not yet reached the Tfps level, and further increasing their frame rates is

[1]Centre Énergie Matériaux Télécommunications, Institut National de la Recherche Scientifique, Université du Québec, 1650 boulevard Lionel-Boulet, Varennes, Québec J3X1P7, Canada. [2]Institut Jean Lamour, Université de Lorraine, Parc de Saurupt CS 50840, Nancy 54011, France. [3]School of Optical and Electronic Information, Huazhong University of Science and Technology, 1037 Luoyu Road, Wuhan 430074 Hubei, China. [4]Present address: Shanghai Institute of Optics and Fine Mechanics, Chinese Academy of Sciences, Shanghai 201800, China. [5]These authors contributed equally: Jingdan Liu, Miguel Marquez. ✉e-mail: jinyang.liang@inrs.ca

fundamentally limited by the electronic bandwidths[20]. Streak cameras —an ultrafast imager converting time to space by pulling photoelectrons with a shearing voltage along the axis perpendicular to the device's entrance slit—can reach an imaging speed of 10 Tfps[21]. Although overcoming the speed limitation, streak cameras are conventionally capable of only one-dimensional imaging[21]. To overcome the drawback in imaging dimensionality, compressed ultrafast photography (CUP)[22–24] adds a single encoding mask on the wide-open entrance port of a streak camera. With the prior information provided by the encoding mask, the spatial information along the shearing direction is allowed to mix with the temporal information in a compressively recorded snapshot. The ensuing image reconstruction recovers the (x,y,t) information. Nonetheless, the produced imaging quality, especially at the Tfps level, can be considerably degraded in the generation and the propagation of photoelectrons by various effects, including the photocathode's thickness, the space-charge effect, and the limited fill factor of the microchannel plate[22]. Meanwhile, space-time coupling in the temporal shearing direction caps the summation of the frame size and the sequence depth in the reconstructed movie, which limits the maximum amount of acquirable information[24]. Finally, temporal shearing induces spatial anisotropy, further reducing the image quality in the reconstructed movie[24].

Active-illumination-based approaches work by imparting the temporal information to various photon tags—such as space, angles, spatial frequencies, and wavelengths—carried in the illumination for 2D imaging at the Tfps level. However, these methods have various limitations. For example, space-division-based techniques[25,26] require the targeted scene to move at a certain velocity to accommodate the sequential arrival of spatially separated probe pulses. Angle-dependent probing is also affected by parallax in each produced frame[27]. The systems relying on spatial frequency division[28] and wavelength division[29] may also face difficulties in scalability in their pattern projection modules and spatial mapping devices. Most importantly, these methods acquire data by compartmenting the focal plane array in either the spatial domain or the spatial frequency domain, which forbids information overlapping. Given the limitations in the sensor's size and the system's optical bandwidth, this focal-plane-division strategy inherently limits the recordable capacity of spatial and temporal information, which usually results in a shallow sequence depth (i.e., the number of frames in each movie).

The limitations in these methods can be lifted using the multi-pattern encoding strategy[16]. Each frame of the scene is encoded with a different pattern at a rate much higher than the sensor's acquisition speed[30]. The captured snapshot thus represents the temporal integration of the spatiotemporally modulated dynamic scene. Then, a compressed sensing-based algorithm is used to reconstruct an ultrafast movie with high quality[31]. As an example, a flutter shutter was implemented to globally block and transmit light in a random sequence during the camera's exposure[32]. This modulation created a more broadband temporal impulse response, which improved the sensing matrix's condition number and hence reconstructed image quality. Teaming up with a multiple-aperture design, this scheme enabled an imaging speed of 200 million fps[33]. However, this global encoding method resulted in a full spatial correlation of the modulation structure imparted on the signal, which limited the compression ratio and hence sequence depth. Thus, ultrafast encoding over each pixel is beneficial from the standpoint of improving reconstruction fidelity[34]. This pixel-wise coded exposure was implemented by using various techniques, such as spatial light modulators (e.g., a digital micromirror device[35,36] and a liquid-crystal-on-silicon device[37]), a translating printed pattern[38,39], and in-pixel memory in the CMOS architecture[40]. However, the imaging speeds enabled by these methods are clamped to several thousand fps by either the pattern refreshing rates of the spatial light modulators[41], the moving speed of the piezo stages, or the readout electronics of the imaging sensor.

Although CUP provides an ultrafast pixel-wise encoding scheme, its operating principle requires simultaneously shearing the scene and the coded aperture. Consequently, pixels across the sensor are encoded with reduced depths, resulting in inferior image reconstruction.

To overcome the limitations in existing methods, here, we report swept coded aperture real-time femtophotography (SCARF), which enables a full pixel-wise encoding depth in single-shot ultrafast imaging by using a single chirped pulse and a modified pulse shaping setup[42]. Leveraging time-spectrum mapping and spectrum-space sweeping, SCARF attaches pixel-wise coded apertures to an ordinary CCD camera at up to 156.3 THz in real time. We demonstrate SCARF in multiple spatial and temporal scales in both reflection and transmission modes. To show SCARF's broad utility, we use it for single-shot real-time imaging of 2D transient light-matter interactions, including ultrafast absorption on a semiconductor and ultrafast demagnetization in a metal alloy.

## Results
### System and principle of SCARF
The SCARF system is shown schematically in Fig. 1a (with an animated illustration in Supplementary Movie 1, as well as component details in Methods). A single linearly chirped laser pulse illuminates a dynamic scene as a continuous probe. Because of its linear chirp, each wavelength in the pulse's bandwidth carries a specific timestamp. After transmitting through the dynamic scene, this probe pulse is recorded in a snapshot by an imaging system modified from a pulse-shaping setup. First, the pulse is imaged by a dispersive 4f imaging system (consisting of lenses L1 and L2 and a grating G1) to a static coded aperture embodied by a pseudo-random binary transmissive mask. The spectral dispersion shears temporal information contained in wavelengths to different positions for spatial encoding by the mask. Then, the pulse is relayed to a CCD camera by another dispersive 4f imaging system (with lenses L3 and L4 and a grating G2) that mirrors the configuration of the first one, which provides the second spectral shearing in the reverse direction. This configuration not only cancels the smearing in the dynamic scene but also sweeps the static coded masks of individual wavelengths (thus at different times). The camera records a compressed snapshot of the dynamic scene's temporal information that is read out by the single probe pulse (details of the forward model are explained in Methods and Supplementary Note 1 and are illustrated in Fig. 1b). After data acquisition, the captured snapshot is input into a compressed sensing-based algorithm that solves a minimization problem to retrieve the (x,y,t) datacube of the dynamic scene (details of image reconstruction and system calibration are explained in Methods, Supplementary Notes 2–3, and Supplementary Fig. 1). SCARF's specifications are summarized in Supplementary Table 1.

SCARF brings in several salient advantages compared to existing techniques in single-shot compressed temporal imaging (summarized here and further explained in Supplementary Notes 4–5, Supplementary Tables 2, 3, and Supplementary Figs. 2, 3). Because of the two symmetric 4f systems, the static coded aperture is swept at a speed of up to $v_s = 1.7 \times 10^9$ m/s, which enables ultrafast pixel-wise spatiotemporal encoding. The sweeping speed also determines SCARF's frame rate by $r = v_s/d$, where d is the binned CCD camera's pixel width in the sweeping direction (see derivation in Supplementary Note 4). This configuration provides encoding rates of up to 156.3 THz to individual pixels on the employed CCD camera, hence enabling single-shot real-time femtophotography. Without using a streak camera, the all-optical data acquisition implemented in SCARF produces superior quality to CUP in the compressively recorded snapshot. Meanwhile, with a compression ratio equal to the sequence depth, SCARF's frame size always equals the sensor size regardless of the duration of the dynamic scene. This paradigm has a stronger information acquisition ability than CUP's sensing model, where the summation of the frame

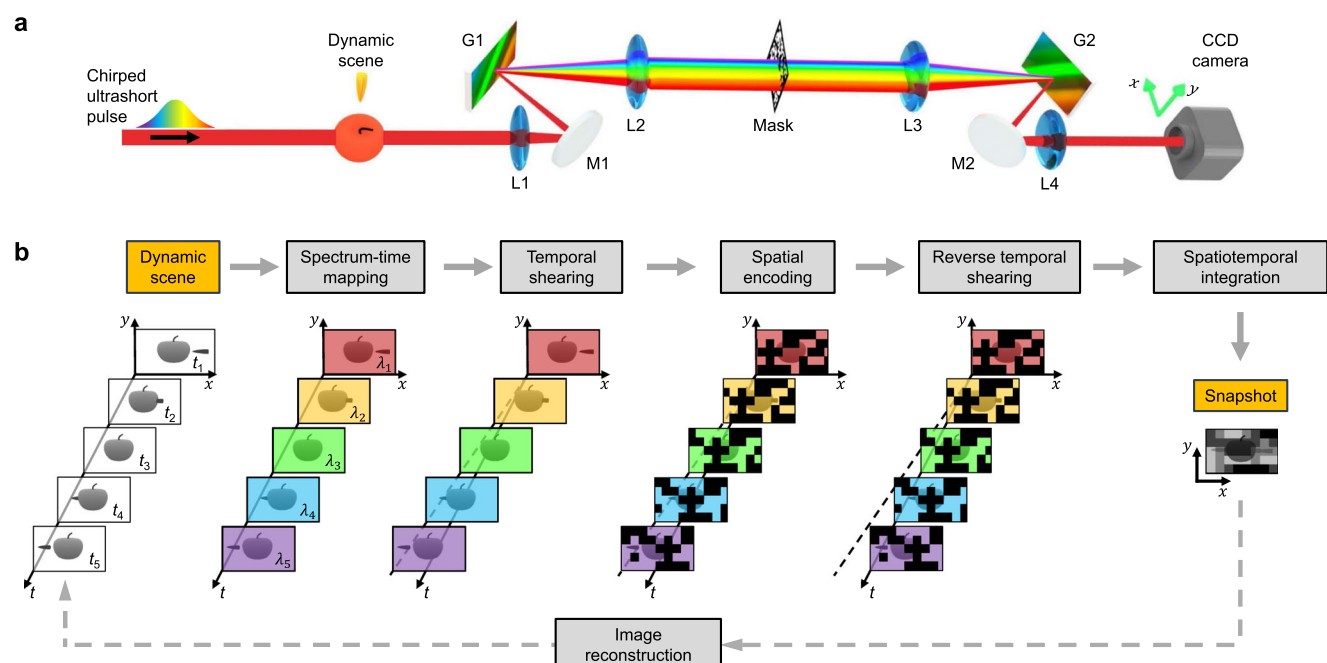

**Fig. 1 | Principle of swept coded aperture real-time femtophotography (SCARF). a** System schematic. CCD Charge-coupled device, G1–G2 Grating, L1–L4 Lens, M1–M2 Mirror. **b** Operation of SCARF with illustrative data. An animated illustration is also provided in Supplementary Movie 1.

size and the sequence depth is restricted by the sensor size. Moreover, SCARF enables encoding each frame with distinct patterns, ensuring that every row of the sensing matrix is non-zero and linearly independent compared to CUP. As a result, SCARF's full row-rank property leads to a trivial null space, which enhances the condition number, reduces sensitivity to noise, and decreases data ambiguity, all of which facilitate a faster, more reliable, and more accurate reconstruction. Finally, in the snapshot, the area with non-zero intensity delineates the region of occurrence of the dynamic scene. This spatial constraint, naturally embedded in the acquired data, facilitates image reconstruction. Detailed explanations and demonstrations are included in Supplementary Note 4 and Supplementary Figs. 2, 3.

## SCARF of single ultrashort pulses transmitting through transparencies

We first prove the concept of SCARF by imaging single chirped pulses transmitting through patterned transparencies at multiple imaging speeds. As shown in Fig. 2a, the chirped pulse was divided by a beam splitter. The reflected component was measured by a second harmonic generation frequency-resolved optical gating (SHG-FROG) device (see details in Supplementary Note 6). The transmitted component passed through a transparency film with a printed pattern. Figure 2b shows five representative frames of a chirped pulse [with a full-width-at-half-maximum (FWHM) duration of 362 fs, as shown in Fig. 2c] passing through a bar pattern imaged by SCARF at 116.3 Tfps. Full sequences of these events (with a sequence depth of 132 frames) are shown in Supplementary Movie 2. As shown in Fig. 2c, the single-shot result well agrees with the scanned SHG-FROG measurements. Finally, the linear relationship between the spectrum and time of the pulse was verified, as shown in Fig. 2d. The details for two other experiments at imaging speeds of 74.9 Tfps and 56.8 Tfps are summarized in Supplementary Note 6, Supplementary Figs. 4, 5, and Supplementary Movies 3, 4.

## SCARF of ultrafast absorption in semiconductor

To show the broad utility of SCARF, we implemented it to monitor two ultrashort phenomena of light-matter interactions in 2D. As the first demonstration, we imaged femtosecond laser-induced ultrafast absorption in a semiconductor (Fig. 3a). A single 40-fs pump pulse

passed through a beam shaping stage before obliquely illuminating a zinc selenide (ZnSe) plate. The high intensity induced by this laser pulse abruptly increased the free carrier density near the surface of this plate, decreasing its transmissivity in tens of femtoseconds[43]. Besides, the oblique incidence led to a non-information faster-than-light propagation of the absorption front on the plate (see the detailed derivation in Supplementary Note 7 and Supplementary Fig. 6). The pump pulse damaged the sample permanently after a single pulse, thus creating non-repeatable transient phenomena.

We studied this phenomenon in two configurations. In the first configurations, the beam shaping stage consisted of an axicon and a focusing lens. By using a 360-µJ pump pulse and an incident angle of $\theta = 35.4°$, we generated an elliptical ring on the ZnSe plate. The event was probed by a 6.5-ps chirped pulse with a normal angle of incidence, corresponding to an imaging speed of 6.5 Tfps. The reconstructed movie of SCARF is shown in Supplementary Movie 5, and eight selected frames are presented in Fig. 3b. The decrease of transmissivity at one vertex of this ellipse immediately breaks in a pair of traces propagating toward opposite directions. These traces, whose time courses delineate the ellipse, eventually converge at the other vertex. Figure 3c presents the normalized average intensity evolution of a five-pixel line centered at the ellipse's upper vertex (marked by the vertical green line in Fig. 3b) with background compensation, which produced the temporal edge spread function (TESF). Taking the derivative of the TESF yielded SCARF's temporal response function (TRF), whose FWHM was quantified to be 226.4 fs.

We also calculated the instantaneous velocities for both traces. As shown in Fig. 3d, the result demonstrates in the $y$ direction, the speeds of both traces start at infinity with opposite directions, then reduce to zero, and finally reaccelerate to infinity with the opposite propagating directions. In contrast, the speeds in the $x$ direction stay at a constant superluminal value of $(5.0 \pm 0.8) \times 10^8$ m/s (mean ± standard deviation). The measured values well match the theory (i.e., $5.2 \times 10^8$ m/s), showing SCARF's excellent ability to track even superluminally moving objects.

In the second configuration, the beam shaping stage contained a cylindrical lens. By using a 40-fs, 124-µJ pump pulse with an incident angle of $\theta = -2.0°$, we generated a line on the ZnSe plate. The absorption front propagated with an apparent velocity of $8.5 \times 10^9$ m/s in the

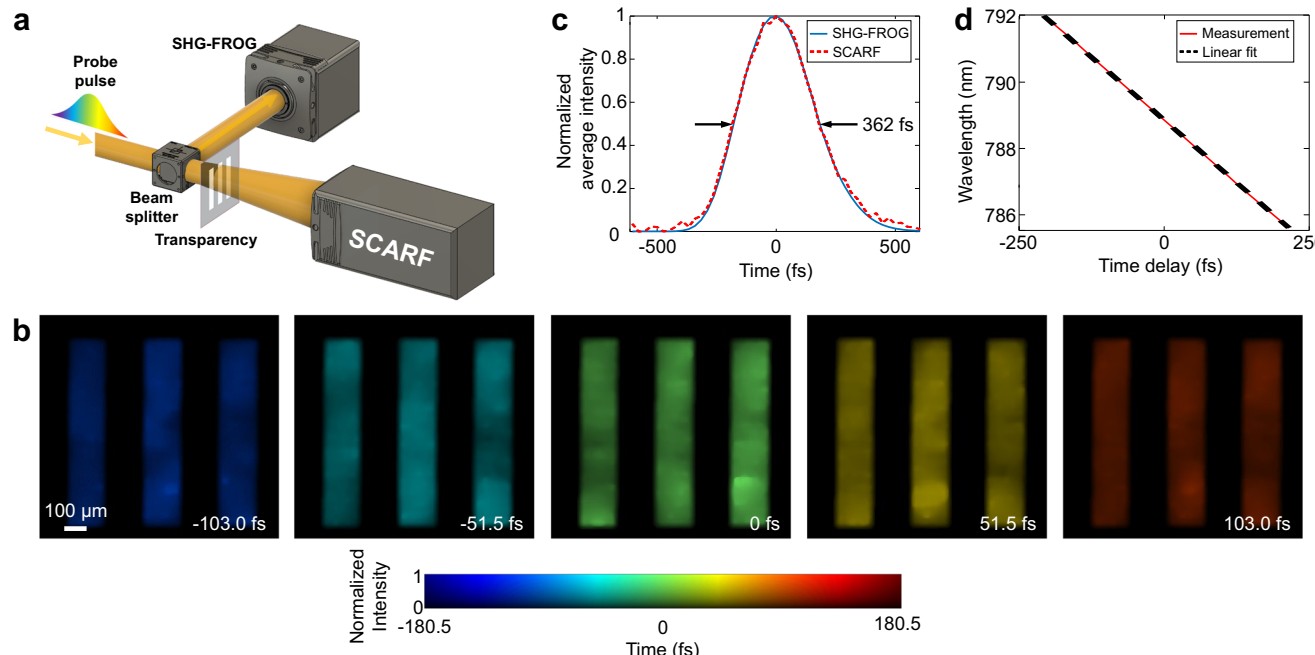

**Fig. 2 | SCARF of a single ultrashort pulse transmitting through a bar transparency. a** Schematic of the experimental setup. SHG-FROG Second harmonic generation frequency-resolved optical gating. **b** Representative frames from the reconstructed movie at 116.3 Tfps. **c** Time course of the normalized average intensity for the reconstructed movie presented in **b**. The measurements using the SHG-FROG technique are also shown as references. **d** Measured relation between the wavelength and time of the chirped pulse with a linear fit.

−x direction. This non-repeatable phenomenon was probed by a 530-fs chirped pulse with a normal angle of incidence, corresponding to an imaging speed of 156.3 Tfps in a single shot. The reconstructed evolution is shown in Supplementary Movie 6, and nine selected frames are presented in Fig. 3e. Figure 3f shows the time course of the normalized averaged intensity in a four-pixel line in the middle of the generated pattern. The TRF was quantified to have an FWHM of 22.4 fs. Considering ZnSe's response time[43], SCARF's temporal response was estimated to be 19.0 fs. Finally, Fig. 3g shows the time course displacement of the absorption front in the middle of the line. The linear-fitted front propagation speed was quantified to be $(8.7 \pm 0.2) \times 10^9$ m/s, showing an excellent agreement with the theory. This experiment demonstrates that SCARF's highest imaging speed can be used to resolve fine spatiotemporal details in light-matter interactions.

## SCARF of ultrafast demagnetization of an alloy film

We imaged ultrafast demagnetization of a pre-magnetized GdFeCo alloy film[44]. As shown in Fig. 4a, a 40-fs, 6.4-μJ pump pulse was loosely focused on this film with an incident angle of 37° to induce ultrafast demagnetization. A 1.2-ps linearly chirped pulse (with +45° linear polarization) probed two selected 2D areas (inset in Fig. 4a) at 19.1 Tfps via a reflection-mode microscope setup. The demagnetization induced a small change in the polarization angle of the probe pulse, which was detected by polarization-resolved SCARF (details are described in Supplementary Note 8 and Supplementary Fig. 7). The full demagnetization process is shown in Supplementary Movie 7. Figure 4b shows the stack-ups of the 2D intensity distribution of the s- and p-polarized light in both Bar 1 and Bar 2. Before the incidence of the pump pulse, the intensities of both channels stay almost equal. The impingement of the pump pulse on the sample increases the intensity of the p-polarized light while decreasing that of the s-polarized light. To quantitatively analyze the results, we plot the time courses of the intensity difference between s- and p-polarized light, as shown in Fig. 4c. The demagnetization times of Bar 1 and Bar 2 were calculated to be $187.5 \pm 18.6$ fs and $186.9 \pm 19.8$ fs, respectively. Both values are in

good agreement with the literature[44] as well as with the multi-shot experiment (see details in Supplementary Note 8 and Supplementary Fig. 7b). The result also reveals that the onset of this change was different by 36.6 fs between these two areas, which was attributed to the oblique incidence of the pump pulse. To quantitatively showcase the advantage of SCARF in this particular case, we averaged the data from these two areas as if spatially resolved imaging was not available. As shown in Supplementary Fig. 7c, the demagnetization time is quantified to be 193.4 fs, which produces an error of 3.9%. Thus, the 2D real-time ultrafast imaging provided by SCARF leads to more informative and more accurate quantification of demagnetization time. Meanwhile, because the demagnetization strength is known to be sensitive to the energy of the pump laser[45,46], single-shot SCARF is immune to the shot-to-shot variation induced by the pump laser's fluctuation (see additional data in Supplementary Fig. 7d).

## Discussion

SCARF advances the frontier of ultrafast optical imaging in both sensing concepts and technical specifications. Its hardware arrangement embodies the multi-pattern encoding paradigm through the ultrafast sweeping of a static coded aperture. The enabled full-sequence encoding in the spatial domain offers a bandwidth of up to 156.3 THz to every pixel on an ordinary CCD camera, which is more than three orders of magnitude greater than the theoretical limit of semiconductor sensors[20]. Moreover, SCARF's image acquisition paradigm is distinguished from existing coded femtophotography techniques by embedding its sensing matrix with attractive mathematical advantages (i.e., full-row rank property and trivial null space), which considerably enhances image reconstruction performance and hence overall sensing ability. Constructed by using off-the-shelf and passive optical components, SCARF is low cost, has low power consumption, and possesses high measurement quality in data acquisition compared to streak-camera-based CUP techniques. Altogether, SCARF separates itself from all existing femtophotography by exhibiting an all-optical

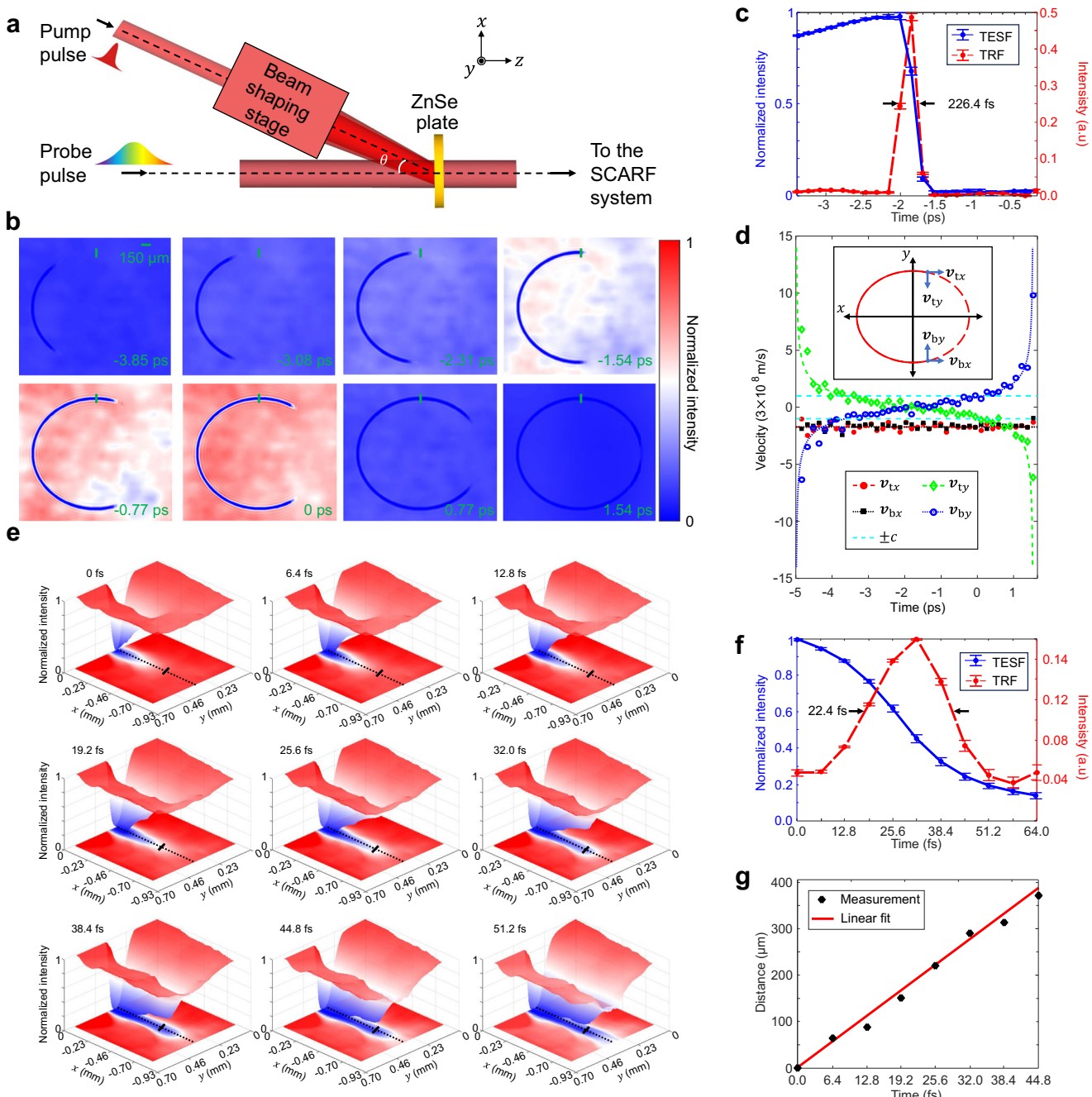

**Fig. 3 | SCARF of ultrafast absorption of shaped laser pulses incident on a zinc selenide (ZnSe) plate. a** Schematic of the experimental setup. **b** Representative frames of the generation of an elliptical absorption pattern imaged at 6.5 Tfps. **c** Temporal response characterization at 6.5 Tfps. TESF, temporal edge spread function [calculated by averaging the intensity time courses of a selected line marked in **b**]. TRF, temporal response function [calculated by taking the derivative of the TESF]. **d** Measured velocities in units of the vacuum speed of light, $c$, of the

top and bottom fronts of absorption (labeled by the markers) along the $x$ and $y$ directions compared to the theoretical predictions (shown as the dashed lines). **e** Representative frames of the generation of a line absorption pattern imaged at 156.3 Tfps. **f** Temporal response characterization at 156.3 Tfps. **g** Measured propagation distance of the absorption front along the $-x$ direction with a linear fit. The marker center and the error bar in **c** and **f** represent the mean value and standard deviation, respectively.

ultrafast real-time imaging modality with tunable imaging speeds (from 6.5 to 156.3 Tfps) and sequence depths (up to 132 frames). These specifications could be further improved by using advanced supercontinuum sources[47], high dynamic range cameras[48], Fourier-domain amplification[49,50], and machine learning image reconstruction[51,52]. Provided the proper sources and sensing devices, SCARF could even be extended to other spectral ranges, such as X-ray and mid-infrared.

SCARF's temporal imaging ability is determined by its temporal resolution rather than its inter-frame time interval. The former describes the image system's response to an impulse in time. The latter, defined as the reciprocal of the imaging speed, expresses the sampling density in time. The relation between these two parameters can be understood in the same way as the spatial resolution and the pixel density[53]. In this work, by using the ultrafast absorption of ZnSe as the temporal impulse, SCARF's temporal response was quantified to

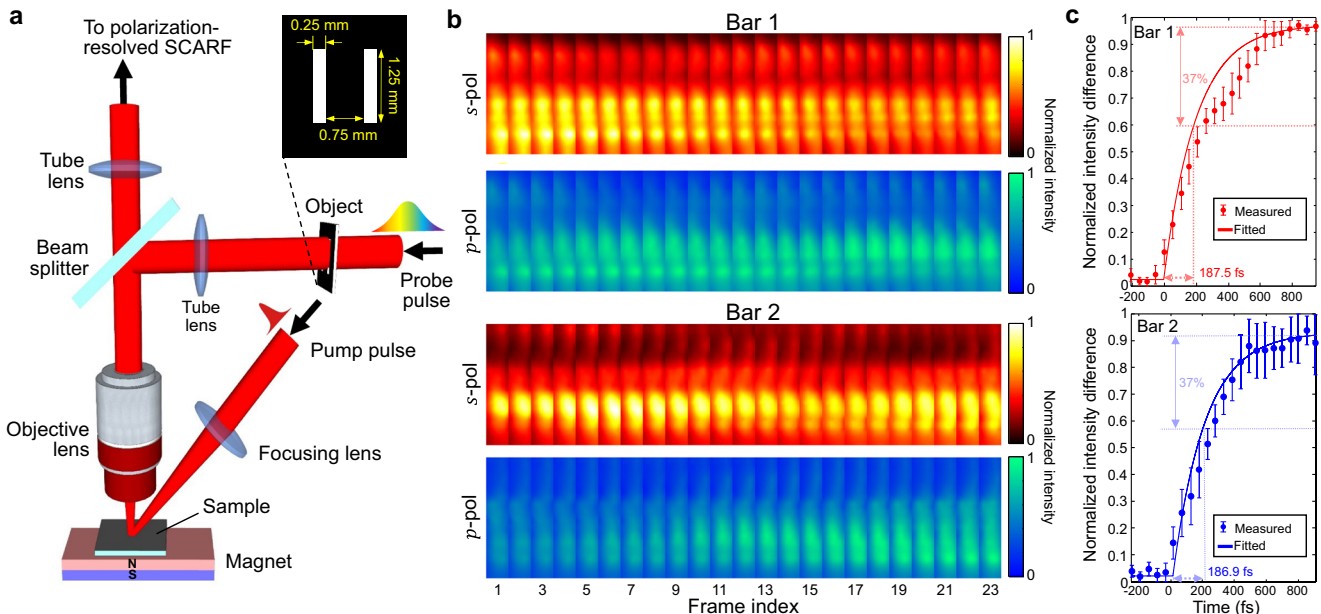

**Fig. 4 | SCARF of ultrafast demagnetization of a GdFeCo thin film. a** Schematic of the experimental setup. The probe pulse passes through an object (see inset) to probe two different areas of the sample simultaneously. **b** Stack-ups of normalized intensity of s- and p-polarization in the two selected areas. **c** Time-resolved normalized intensity difference between the s- and p-polarization light for Bar 1 (top) and Bar 2 (bottom). The marker center and the error bar in **c** represent the mean value and the standard deviation, respectively.

have an FWHM of 226.4 fs for 6.5 Tfps (inter-frame time interval of 153.8 fs) and 19.0 fs for 156.3 Tfps (inter-frame time interval of 6.4 fs). Future theoretical studies could establish the relation between SCARF's temporal resolution and its other parameters to optimize the system design.

SCARF allows efficient observation of transient events with high imaging quality. Detailed in Methods, the deployed light source provided pulse energy of up to 1.6 mJ. For all the experiments in this work, the maximum probe pulse energy of this active illumination saturated the CCD camera in the SCARF system. Thus, optimized signal-to-noise ratios in data acquisition could always be obtained by tuning optical attenuation, which facilitated the reconstruction of high-quality images. Meanwhile, SCARF's tunability was leveraged to adapt the system's specifications to the experiments. The generation of the targeted events was synchronized with SCARF's data acquisition. According to the estimated durations of these events, the optimal combination of the imaging speed and the sequence depth was searched for each experiment with the criterion of clearly observing the event's evolution while ensuring discernible differences in neighboring frames recording the event's evolution. In this way, temporal oversampling, hence information redundancy, was avoided in SCARF.

SCARF expands the application scope of the classic pulse shaper to single-shot ultrafast imaging. Starting from a 4*f* setup[54] (i.e., G1, L2, L3, and G2 in Fig. 1a), SCARF's hardware is built with two modifications—using a pseudo-random binary mask for amplitude modulation and adding two additional lenses (i.e., L1 and L4 in Fig. 1a) to perform optical Fourier transformation before and after the two gratings. This arrangement converts the pulse shaper to two cascaded dispersive 4*f* imaging systems interlinked by the encoding mask. Its symmetric configuration naturally enables spectral shearing in opposite directions without the need for synchronization. The Fourier plane of a conventional pulse shaping setup, which is now occupied by the encoding mask, still manipulates individual wavelengths of a broadband laser pulse[42]. However, different from the conventional operation, the incorporation of the lens L1 and the linearly chirped pulse brings the encoding of the sequential time information with different sections of a mask in the spatial domain.

Moreover, the implementation of the compressed sensing model allows the mixture of spectral and spatial information, nullifying the line-focusing necessity in a conventional 4*f* setup. Finally, by exploiting the time-spectrum-space conversion, the SCARF system converts the ultrashort timescale of a chirped ultrafast pulse to ultrafast sweeping of the coded aperture. In summary, SCARF inherits the advantages from the classic pulse shaper and augments its optical processing ability, particularly for observing transient single events.

As a generic and economical imaging modality, SCARF has promising applications in ultrafast science. Of particular relevance are the two light-interaction phenomena studied in this work. First, single-shot SCARF of ultrafast absorption on ZnSe could contribute to the study of ultrafast carrier dynamics in semiconductor thin films and 2D materials in their sub-bandgap region[55]. SCARF could probe the spatial distribution of the ultrafast transient absorption by the excited states of diffusing carriers. This investigation will contribute to overcoming the intrinsic bandgap limitations in modulators and photodetectors using excited carrier states[56]. Meanwhile, single-shot SCARF of ultrafast demagnetization of metal alloys may open a new route for studying ultrafast magnetic switching for possible future applications of magnetic storage devices[57,58]. Current time-resolved methods[59,60] require using numerous probe pulses at different delays and rely on the reproducibility of the phenomenon to procure the dynamics. In contrast, to our knowledge, SCARF marks the debut of single-shot 2D optical imaging of ultrafast demagnetization. It could be applied to imaging longitudinal ultrafast all-optical switching of various magnetic thin films[59,61,62], which will provide experimental evidence of the maximum reliable switching rate for the next-generation magnetic storage[63]. Coupling SCARF into wide-field super-resolution microscopy could be particularly valuable for the study of domain dynamics during ultrafast demagnetization and switching. Other potential applications of SCARF include single-shot 2D probing of opto-mechanical motion of micro/nano-sized objects[64], irreversible chemical reaction dynamics of organic crystals[65], and evolution dynamics of plasma wakes in a laser wake-field accelerator[66].

## Methods

### Operating principle of SCARF

SCARF's data acquisition can be expressed by five successive operations (illustrated in Fig. 1b and derived in Supplementary Note 1). First, time-spectrum mapping (denoted by **M**) is executed when the single chirped pulse probes the dynamic scene, storing temporal information at different wavelengths. Then, the dispersion induced in the first dispersive 4*f* system enables spectral shearing of the dynamics scene (denoted by **S**), followed by spatial encoding by the pseudo-random binary transmissive mask (denoted by **C**). Afterward, the second dispersive 4*f* system induces another spectral shearing in the reverse direction (denoted by **S′**). Finally, the spatially encoded dynamic scene experiences spatiotemporal integration on the CCD camera (i.e., spatially integrating over each pixel and temporally integrating over the exposure time; denoted by **T**). In this way, the captured snapshot, $E[m,n]$, is linked with the transmittance modulated by the dynamic scene $a(x,y,t)$ by

$$E[m,n] = \mathbf{O}a(x,y,t) \tag{1}$$

where $m$ and $n$ are the pixel indices of the CCD camera. The operator $\mathbf{O} = \mathbf{TS'CSM}$.

In the ensuing image reconstruction, $E[m,n]$ is input to an algorithm developed from the plug-and-play alternating direction method of multipliers (PnP-ADMM) framework[67] (details of derivation are shown in Supplementary Note 2). Leveraging the spatiotemporal sparsity of the dynamic scene and the prior knowledge of each operator, $a(x,y,t)$ can be retrieved by solving the minimization problem of

$$\hat{\mathbf{a}} = \underset{\mathbf{a}\in\mathscr{A}}{\mathrm{argmin}}\frac{1}{2}||\mathbf{Oa} - \mathbf{E}||_2^2 + \mathrm{R}(\mathbf{a}) + \mathrm{I}_+(\mathbf{a}) \tag{2}$$

Here, $\mathscr{A}$ represents a set of solutions that satisfy the spatial constraint. **a** is the discrete version of $a(x,y,t)$. $||\cdot||_2$ represents the $l_2$ norm. $\frac{1}{2}||\mathbf{Oa}-\mathbf{E}||_2^2$ is the fidelity term representing the similarity between the measurement and the estimated result. R$(\cdot)$ is the implicit regularizer that promotes sparsity in the dynamic scene[68,69] (further explained in Supplementary Note 2). $\mathrm{I}_+(\cdot)$ represents a non-negative intensity constraint.

### Details on equipment and sample preparation

The components in the SCARF system (Fig. 1a) include four 100 mm-focal-lengths lenses (L1 and L4, LA1509, Thorlabs; L2 and L3, LA1050, Thorlabs), two 1200 line/mm gratings (G1 and G2, GR25-1208, Thorlabs), one static pseudo-random binary transmissive mask (HTA Photomask, 80 μm ×80 μm encoding pixel's size), and a CCD camera (GS3-U3-41C6NIR-C, FLIR, 2048×2048 pixels).

The illumination of the SCARF system was provided by a femtosecond Titanium-Sapphire laser amplifier at the multi-kHz beamline of the Advanced Laser Light Source (ALLS) at the Centre Énergie Matériaux Télécommunications, Institut National de la Recherche Scientifique, Univeristé du Québec. The output pulses have a central wavelength of 780 nm, a pulse energy of 1.6 mJ, a pulse width of 40 fs, and a bandwidth of 26 nm. A two grating-based pulse stretcher was used to generate linearly chirped pulses used in the experiments of this work. All these pulses output from the laser pass through two bandpass filters (LD01-785/10-25 and LL01-810-25, Semrock) so that the durations of the generated probe pulses match those of the dynamic events.

In the dynamic absorption experiment (Fig. 3a), a 1-inch ZnSe plate (WG71050, Thorlabs) was used. An axicon (130-0278, Eksma Optics) and a 150-mm-focal length focusing lens (LA1433-B, Thorlabs) were used to generate the elliptical ring. The focusing lens was placed 360 mm away from the axicon. A 700-mm-focal-length cylindrical lens (LJ1836L1-B, Thorlabs) was used to generate the line.

In the transient demagnetization experiment (Fig. 4a), a GdFeCo alloy thin film was used as the sample. This multi-layer film was arranged by glass substrate/Ta(3 nm)/Cu(5 nm)/GdFeCo(20 nm)/Cu(5 nm)/Al(2.5 nm). Other components included an objective lens (MY20X-824, Mitutoyo), a beam splitter (BSW27, Thorlabs), a focusing lens (LA1433-B, Thorlabs), two tube lenses (AC254-200-B, Thorlabs). A mask of two bars (0.25 mm × 1.25 mm in size and 0.75-mm separation) was placed 200 mm away from the tube lens in the illumination beam path.

### Summary of key system parameters

SCARF's field of view depends on the CCD camera's sensor size and the system's overall magnification ratio. For the experiments conducted in this work, the overall magnification ratio was one except for the transient demagnetization experiment (see Fig. 4). There, the use of an objective lens made the overall magnification ratio to be 20 ×. Moreover, the spatial resolution was measured by quantifying the edge spread function of a fine spatial feature in the reconstructed videos. Moreover, 2 × 2 pixel binning was implemented to improve the quality of the acquired snapshot. Finally, derived in Supplementary Notes 1 and 4, SCARF's imaging speed is determined by $r = f_2\alpha/d\beta$, where $f_2$ is the focal length of Lens 2, $\alpha$ is the angular dispersion of grating G1, $d$ is the binned pixel size of the deployed camera, and $\beta$ is the linear time-spectrum mapping parameter. As shown in Section "System and principle of SCARF", it can be derived by using the definition of the sweeping velocity $v_s = f_2\alpha/\beta$. Technical specifications of the SCARF system for each experiment are summarized in Supplementary Table 1.

## Data availability

All data supporting the results of this study are available within the paper and its Supplementary Information. Representative raw data generated in this study have been deposited in the Figshare data repository (https://doi.org/10.6084/m9.figshare.24407194). Additional data are available for research purposes from the corresponding author upon request.

## Code availability

The image reconstruction and processing algorithms are described in detail in Supplementary Information. The reconstruction software uses a freely available universal ADMM algorithm (https://web.stanford.edu/~boyd/papers/admm/) and BM3D software (https://webpages.tuni.fi/foi/GCF-BM3D/). The customized reconstruction codes are available from the corresponding author upon request.

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

## Acknowledgements

The authors thank Prof. Donna Strickland from the University of Waterloo for the fruitful discussion. This work was supported in part by Natural Sciences and Engineering Research Council of Canada (ALLRP-551076-20, ALLRP-549833-2020, CRDPJ-532304-18, RGPAS-507845-2017, RGPIN-2017-05959) (J.Liang), Canada Research Chairs Program (CRC-2022-00119) (J.Liang), Canada Foundation for Innovation and Ministère de l'Économie et de l'Innovation–Gouvernement du Québec (37146) (J.Liang), New Frontier in Research Fund (NFRFE-2020-00267) (J.Liang), Fonds de recherche du Québec–Nature et technologies (2019-NC-252960, 203345 - Centre d'optique, photonique et lasers) (J.Liang), and Fonds de Recherche du Québec - Santé (267406, 280229) (J.Liang).

## Author contributions

J.Liu and H.I. built the SCARF system. M.M., Y.L., and X.L. developed the SCARF's image reconstruction algorithm. M.M., J.Liu, H.I., P.L., and K.L. performed the experiments and analyzed the data. M.H., S.M., and G.M. prepared the GdFeCo sample. J.Liang, M.M., H.I., and Z.L. developed the SCARF theory. J.Liu, M.M., and J.Liang drafted the manuscript. J.Liang and F.L. initiated the project. J.Liang proposed the conceptual system and experimental design and supervised the project. All authors revised the manuscript.

## Competing interests

The authors declare no competing interests.
