## [Peer Review File · Nature Communications]

Swept coded aperture real-time femtophotographyEditorial Note: This manuscript has been previously reviewed at another journal that is not operating a transparent peer review scheme. This document only contains reviewer comments and rebuttal letters for versions considered at Nature Communications.

REVIEWER COMMENTS

Reviewer #1 (Remarks to the Author):

The paper describes an optical arrangement (SCARF) that enables ultrafast videography by merging the CUP method with STAMP. The paper has several interesting aspects and I truly appreciate the technical aspects to the optical configuration and the hard work behind the development and experiments.

However, when reading the updated version of the paper, I still believe the authors claim progress that they do not verify or demonstrate experimentally. In the abstract it is written: "Existing techniques are either strained by the trade-off between fields of view and sequence depths or limited by spatial resolution and imaging speeds. Here, we overcome these limitations...". Yet in the paper the obtained FOV, sequence depth and spatial resolution are the same or even worse/lower than existing ultrafast imaging techniques. The imaging speed is reported to be a factor of 2 better than its competition, which is impressive and noteworthy, yet the authors do not report their temporal resolution, so the reader cannot easily judge whether the boosted imaging speed can be exploited. The authors still assume that the readers can assess the technical improvements based on the technical description.

All in all, the paper presents a sound technology but it does not provide experimental evidence that it does what is claimed and I can therefore not recommend publication. I have listed points the authors need to address before submitting the manuscript again.

1. Temporal resolution

1.1 The paper states an imaging speed of 156.3 Tfps, which is impressive and outstanding. The authors also explain well how they define this imaging speed (lines 124-126). However, imaging speed does not equal temporal resolution (in the same way that pixel density does not equal spatial resolution) and it is really the temporal resolution that is relevant and what the authors need to report. I have no doubt that the paper outperforms its competition in terms of temporal resolution but given the topic and claims of the paper, the authors need to discuss and estimate it. If the authors cannot estimate it, they at least need to clarify for the reader that the imaging speed is not always equal to the temporal resolution of the system (see point 1.2).

1.2 To be able to make use of an extreme imaging speed of 156.3 Tfps when studying spatially dynamic objects (i.e. moving objects/phenomena), the system needs to have a matching spatial resolution. For example, if the spatial resolution equals 33 μm (experiment in Supp. Fig. 4), the temporal resolution needs not to be greater than 110 fs for non-superluminal objects, otherwise the data becomes oversampled. For superluminal objects on the other hand, the speed by which the object evolves needs to be around 17 times the speed of light for motion occurring during 6.4 fs to be detected (with the same spatial resolution). This explains why the authors are able to visualize the progressing absorption in Supp. Fig. 4. Please note that I am not trying to say that the authors are wrong in their results, just that this information and aspect of ultrafast videography is missing and is, in my view, very important and I encourage the authors to include such a discussion to avoid misunderstandings.

2. Measurement through transparencies

2.1 I still believe that the static samples do not add much to the paper, with respect to the four parameters the authors build their case on: FOV, imaging speed, spatial resolution and sequence depth. If the authors have measurements performed on a significantly shorter laser pulse or at its highest imaging speed, the results would be relevant and provide support to the claimed

advancement.

2.2. The stated FOV in Supp. Table 1 (11x11 mm) is inconsistent with the presented FOV ($\sim 2 \times 2$ mm).

3. Kerr medium experiment

3.1 I do not understand the purpose of this experiment, it does not seem to demonstrate any of the aspects the authors bring out as important. Firstly, it is recorded at 1.6 Tfps, i.e. at $\sim 1\%$ of the maximum imaging speed of 156 Tfps, without motivating the reduced speed, especially considering the speed by which the object is moving. Secondly, the reduced imaging speed leads to a loss in spatial resolution as light travels ~ 91 μm during one frame. It therefore becomes near impossible for the authors to make any claim about any improvements in spatial resolution offered by the system. Thirdly, the presented FOV is $\sim 1 \times 2$ mm, which is lower than streak camera-based configurations it is said to outperform. The FOV is also stated to be 11x11 mm in Supp. Table 1, which is not consistent with the presented results. Finally, it is composed out of 12 frames, so the sequence depth is two orders of magnitude lower than CUSP. The authors need to rethink and rewrite the section concerning the Kerr medium so the reader understands its purpose. Instead of demonstrating how well SCARF can visualize ultrafast events, the authors instead become forced to explain it fails to do so and why the pulse appears to be 24 times longer (lines 198-201). Once again, I am not trying to say that the authors are wrong, I just do not see what they are trying to relay.

4. Ultrafast demagnetization experiment

4.1 I am concerned about the conclusions drawn from the measurement presented in Fig. 5. The data points in Fig. 5c are extremely scattered and I have doubts about the accuracy of the extracted trend. Can the authors confirm the findings by e.g. repeated measurements?

5. Ultrafast absorption on a ZnSe plate

5.1 It puzzles me why this experiment is included only in the supplementary material, especially since it is the only experiment supporting the claim of achieving 156.3 Tfps. This seems not very thought-through. Move the result to the main manuscript, possibly replacing either the experiment on the transparencies or the Kerr medium, since neither of those results support the claims of the paper convincingly.

5.2 See point 1.2, I think the discussion I try to highlight there needs to be included for a reader to understand in which way the spatial resolution is a restricting factor.

5.3 Video 6, which is recorded at 156.3 Tfps, have either a high pixel binning or is digitally zoomed in. According to the paper, pixel binning reduces the frame rate (line 125) so please confirm that the data is not binned.

5.4 The stated FOV in Supp. Table 1 (11x11 mm) is inconsistent with the presented FOV ($\sim 1 \times 1$ mm).

Responses to Reviewer

We sincerely appreciate the reviewer for the thorough review of our work and the insightful and constructive comments, which are helpful for greatly improving the quality of our manuscript. In the following, we provide point-by-point responses. The changes in the revised manuscript are highlighted in red.

[Comment 0]

The paper describes an optical arrangement (SCARF) that enables ultrafast videography by merging the CUP method with STAMP. The paper has several interesting aspects and I truly appreciate the technical aspects to the optical configuration and the hard work behind the development and experiments.

However, when reading the updated version of the paper, I still believe the authors claim progress that they do not verify or demonstrate experimentally. In the abstract it is written: “Existing techniques are either strained by the trade-off between fields of view and sequence depths or limited by spatial resolution and imaging speeds. Here, we overcome these limitations...”. Yet in the paper the obtained FOV, sequence depth and spatial resolution are the same or even worse/lower than existing ultrafast imaging techniques. The imaging speed is reported to be a factor of 2 better than its competition, which is impressive and noteworthy, yet the authors do not report their temporal resolution, so the reader cannot easily judge whether the boosted imaging speed can be exploited. The authors still assume that the readers can assess the technical improvements based on the technical description.

All in all, the paper presents a sound technology but it does not provide experimental evidence that it does what is claimed and I can therefore not recommend publication. I have listed points the authors need to address before submitting the manuscript again.

[Response 0]

We thank the reviewer for his/her/their acknowledgments of the soundness of technology and our effort in the development and experiment. Following the reviewer’s suggestions, we have fully addressed each comment and revised the manuscript accordingly. The major changes in the revised manuscript are summarized in the following six points:

- (1) We have revised the two sentences in Abstract mentioned by the reviewer as well as all related text in the revised manuscript. We realized that the wording may have caused the confusion. We had no intention to claim the largest field of view (FOV) or sequence depth. We wanted to say that sensing models in existing femtophotography approaches, in particular CUP, impose a limit on the acquirable information in the frame size and/or the sequence depth. In addition, we have no intention to claim that SCARF has the best spatial resolution. We simply wanted to say that existing femtophotography approaches, in particular CUP, have limited image quality from the device deployed in data acquisition. In the revised manuscript, the wording of both claims has been revised, and the working principles and limitations of CUP have now been explicitly mentioned in Introduction (see Lines 50–64 on Pages 2–3).
- (2) We have analyzed the temporal resolution of experimental data and provided explanations. We have also explicitly stated that the SCARF’s imaging performance is limited by the temporal resolution, not by the interframe time interval. Please see the details in **Response 1**.
- (3) From the perspective of information theory, we have analyzed how SCARF distinguishes from CUP in the sensing paradigm and how this difference brings more robustness, higher numerical stability, and better quality in image reconstruction. We have provided demonstrations that SCARF, different from CUP, does not compromise its frame size to accommodate the sequence depth. These distinctions show SCARF is *not* a mere combination of CUP with STAMP. Nonetheless, we would like to reiterate that the aim of this manuscript is not to provide a head-to-head competition with CUP. Please see **Response 2** for details.
- (4) We have explained and revised the description of Fig. 2 and the claims of FOV. We also removed all data/results of Kerr-medium imaging. Please refer to the details in **Responses 2–3**.
- (5) Regarding the single-shot real-time imaging of ultrafast demagnetization, we have conducted additional experiments, improved the results, and verified the findings. Please see details in **Response 4**.
- (6) We moved the 156.3-Tfps experiment of the transient absorption of a ZnSe plate from Supplementary Materials to Main Text. Please see details in **Response 5**.

[Comment 1]

1. Temporal resolution

1.1 The paper states an imaging speed of 156.3 Tfps, which is impressive and outstanding. The authors also explain well how they define this imaging speed (lines 124-126). However, imaging speed does not equal temporal resolution (in the same way that pixel density does not equal spatial resolution) and it is really the temporal resolution that is relevant and what the authors need to report. I have no doubt that the paper outperforms its competition in terms of temporal resolution but given the topic and claims of the paper, the authors need to discuss and estimate it. If the authors cannot estimate it, they at least need to clarify for the reader that the imaging speed is not always equal to the temporal resolution of the system (see point 1.2).

1.2 To be able to make use of an extreme imaging speed of 156.3 Tfps when studying spatially dynamic objects (i.e. moving objects/phenomena), the system needs to have a matching spatial resolution. For example, if the spatial resolution equals 33 μm (experiment in Supp. Fig. 4), the temporal resolution needs not to be greater than 110 fs for non-superluminal objects, otherwise the data becomes oversampled. For superluminal objects on the other hand, the speed by which the object evolves needs to be around 17 times the speed of light for motion occurring during 6.4 fs to be detected (with the same spatial resolution). This explains why the authors are able to visualize the progressing absorption in Supp. Fig. 4. Please note that I am not trying to say that the authors are wrong in their results, just that this information and aspect of ultrafast videography is missing and is, in my view, very important and I encourage the authors to include such a discussion to avoid misunderstandings.

[Response 1]

We completely agree with the reviewer that the time interval between adjacent frames (i.e., the reciprocal of the imaging speed) is different from the temporal resolution of the system. Temporal resolution is an important parameter to evaluate the system's performance. In the revised document, we have added the following content to clarify this point.

First, we have analyzed SCARF's temporal resolution for the imaging of ultrafast absorption on a ZnSe plate. From the literature, the decrease of transmittance induced by the femtosecond laser pulse occurs in tens of femtoseconds [Ref. 42 in the original manuscript; now Ref. 45 in the revised manuscript]. This temporal impulse allowed us to characterize the SCARF's temporal

resolution to be 226.4 fs for 6.5-Tfps imaging and 19.0 fs for 156.3-Tfps imaging. These new results are added as Figs. 3c and f as well as Lines 187–192 and 199–210 in Main Text.

Moreover, we have added a discussion in Section 3 (see Lines 260–269) in Main Text to further distinguish interframe time interval from temporal resolution. First, we have explicitly stated the difference between the temporal resolution and the interframe time interval by using the analogy of pixel density and spatial resolution as suggested by the reviewer. We have clarified that the temporal resolution, which is always larger than the interframe time interval, determines SCARF's temporal imaging ability. Second, we have stated the requirement of a temporal impulse in the quantification of temporal resolution and discussed how the characteristics of transient phenomena determine whether we could obtain the temporal resolution for each experimental configuration used in this work.

Here, we would like to elaborate on the relationship between spatial resolution and imaging speed. Although we generally agree with the reviewer that spatial resolution is an important factor in selecting the system's imaging speed, there exist various scenarios in which these two specifications are not tightly linked. For the moving-object case that the reviewer suggested, a high imaging speed is preferred to procure images with minimal temporal blur. An illustrative example is shown in Fig. R1a. In particular, a single point (i.e., a spatial delta function) moves horizontally at the speed of light c . To visualize the motion, the object needs to move at least one pixel (shown as one grid in Fig. R1). Due to the diffraction limit, this point is imaged as a circular point spread function (PSF) with a diameter of 3 pixels, which illustrates the spatial resolution. If we denote the pixel width as d , then the minimum interframe time interval to resolve the motion (i.e., the maximum effective imaging speed) should be set to $t = d/c$. Figure R1b shows the result at this imaging speed. In contrast, Fig. R1c shows an imaging speed determined by the spatial resolution. With a longer inter-frame time interval (hence a longer exposure time and a lower imaging speed), the produced images have a stronger temporal blurring effect. In Fig. R1c, this effect is illustrated by the oval shape stretched from the circular PSF. Thus, a higher imaging speed can produce better image quality in diffraction-limited imaging of moving objects.

Fig. R1. Effects of imaging speed on diffraction-limited images. (a) Ground truth. (b) Diffraction-limited high-speed imaging. (c) Diffraction-limited low-speed imaging.

For light dynamics occurring on static objects, it is also possible that the spatial resolution is not a restrictive factor. In fact, a sufficiently high imaging speed is key to resolving temporal features in the light dynamics in microscopy (e.g., ultrafast demagnetization imaging in this work), mesoscopy (e.g., fluorescence lifetime imaging for glioma detection [R1]), and macroscopy (e.g., time-of-flight LIDAR [R2]). In these cases, the required interframe time interval (hence temporal resolution) always needs to be short enough to resolve the intensity changes. In Ref. [R2], a 400-ps pulse (corresponding to 120 mm in space) was used as the illumination of this time-of-flight LIDAR system. Nonetheless, using an effective imaging speed of 160 billion frames per second, this work produced a ~ 2 -mm depth resolution, much 32 times finer than the resolution determined by the pulse duration.

[Comment 2]

2. Measurement through transparencies

2.1 I still believe that the static samples do not add much to the paper, with respect to the four parameters the authors build their case on: FOV, imaging speed, spatial resolution and sequence depth. If the authors have measurements performed on a significantly shorter laser pulse or at its

highest imaging speed, the results would be relevant and provide support to the claimed advancement.

2.2 The stated FOV in Supp. Table 1 (11x11 mm) is inconsistent with the presented FOV (~2x2 mm).

[Response 2]

Although we understand the reviewer's reasoning and concerns, we still think that the measurements with static transparencies are still valuable for two reasons. First, this experiment serves as the proof-of-concept demonstration of the SCARF's ability to record ultrafast events at an imaging speed of >100 Tfps. Second, this experiment serves as a guide to ultrafast imaging of demagnetization (Fig. 5 in Main Text), which also involves intensity dynamics occurring on a static sample. Thus, this measurement is necessary for a cogent and flowy presentation.

Nonetheless, following the reviewer's suggestion, we decided to remove most panels in this figure (Figs. 2b–c, e–f, and the associated description in the original manuscript) into Supplementary Note 6 and Supplementary Fig. 5. Only Figs. 2a and d are retained to show the >100 Tfps imaging ability. To ensure the completeness of the presentation, we added a schematic of the experiment setup and moved Supplementary Fig. 3j in the original manuscript to Main Text as the new Fig. 2d in the revised manuscript. The related text in Section 2.2 has also been updated.

In addition, we would like to point out that 11 mm × 11 mm is the largest possible FOV determined by the sensor size. We realize that this number is not used in the experiment. Thus, following the reviewer's suggestion, we have changed all numbers in Supplementary Table 1 to better reflect the actual experimental conditions.

Finally, we have added more analysis to show the difference between SCARF and CUP sensing paradigms. First, we have shown how SCARF exceeds CUP in the acquisition ability of temporal information in scenarios where the transient events occur close to the FOV's borders, as shown in Supplementary Fig. 3a. Second, we have demonstrated how the SCARF sensing matrix's mathematical properties promote the enhancement in image reconstruction, higher numerical stability, robustness, and convergence speed. These results are included in Supplementary Figs. 3b–e. Description of these new data are included in Lines 140–150, Lines 247–255, and Supplementary Note 4 in the revised manuscript.

[Comment 3]

3. Kerr medium experiment

3.1 I do not understand the purpose of this experiment, it does not seem to demonstrate any of the aspects the authors bring out as important. Firstly, it is recorded at 1.6 Tfps, i.e. at ~1% of the maximum imaging speed of 156 Tfps, without motivating the reduced speed, especially considering the speed by which the object is moving. Secondly, the reduced imaging speed leads to a loss in spatial resolution as light travels ~91 μm during one frame. It therefore becomes near impossible for the authors to make any claim about any improvements in spatial resolution offered by the system. Thirdly, the presented FOV is ~1x2 mm, which is lower than streak camera-based configurations it is said to outperform. The FOV is also stated to be 11x11 mm in Supp. Table 1, which is not consistent with the presented results. Finally, it is composed out of 12 frames, so the sequence depth is two orders of magnitude lower than CUSP. The authors need to rethink and rewrite the section concerning the Kerr medium so the reader understands its purpose. Instead of demonstrating how well SCARF can visualize ultrafast events, the authors instead become forced to explain it fails to do so and why the pulse appears to be 24 times longer (lines 198-201). Once again, I am not trying to say that the authors are wrong, I just do not see what they are trying to relay.

[Response 3]

The main purpose of this experiment is to demonstrate SCARF's ultrafast imaging ability with the contrast of refractive index. Meanwhile, we aim to demonstrate SCARF's multi-scale temporal imaging ability by exploring different binning. We agree with the reviewer that the specifications used in this experiment are relatively inferior to others presented in this work. Therefore, we have removed all data from this experiment (i.e., Fig. 4 and Supplementary Fig. 5) as well as the related text (i.e., Section 2.4 and Supplementary Note 8) in the original manuscript.

[Comment 4]

4. Ultrafast demagnetization experiment

4.1 I am concerned about the conclusions drawn from the measurement presented in Fig. 5. The data points in Fig. 5c are extremely scattered and I have doubts about the accuracy of the extracted trend. Can the authors confirm the findings by e.g. repeated measurements?

[Response 4]

Yes, we have repeated the ultrafast demagnetization experiments and improved the reconstruction quality. A new result is shown in Figs. 4b–c in the revised manuscript. Another dataset is included in Supplementary Note 8 and Supplementary Fig. 7d in the revised manuscript. Both results show similar trends and demagnetization times.

[Comment 5]

5. Ultrafast absorption on a ZnSe plate

5.1 It puzzles me why this experiment is included only in the supplementary material, especially since it is the only experiment supporting the claim of achieving 156.3 Tfps. This seems not very thought-through. Move the result to the main manuscript, possibly replacing either the experiment on the transparencies or the Kerr medium, since neither of those results support the claims of the paper convincingly.

5.2 See point 1.2, I think the discussion I try to highlight there needs to be included for a reader to understand in which way the spatial resolution is a restricting factor.

5.3 Video 6, which is recorded at 156.3 Tfps, have either a high pixel binning or is digitally zoomed in. According to the paper, pixel binning reduces the frame rate (line 125) so please confirm that the data is not binned.

5.4 The stated FOV in Supp. Table 1 (11x11 mm) is inconsistent with the presented FOV (~1x1 mm).

[Response 5]

We thank the reviewer for taking a detailed look at our manuscript. In a previous round of review in another journal, we included the results of this 156.3-Tfps experiment in Supplementary Materials to follow another reviewer’s suggestion. Despite our desire to place it in Main Text, we also tried carefully to satisfy the taste of every reviewer, especially after four rounds of review.

We agree with the reviewer that this result should be emphasized. Therefore, in the revised manuscript, we have moved Supplementary Figs. 4a, c, and d as well as the associated text in the original manuscript to Main Text. These changes are shown in Fig. 3, Supplementary Fig. 8, and Section 2.3 in the revised manuscript.

Regarding the role of spatial resolution, please refer to our reasoning in Response 1 about why it is not a restricting factor for ultrafast imaging. For this particular experiment, a high imaging speed facilitates the capture of the moving absorption front with minimal temporal blur.

Moreover, we confirm that the data in Video 6 do not have high pixel binning. For this experiment, the dynamic event occurs in a narrow line, therefore, we selected a relatively small FOV (i.e., $0.78 \text{ mm} \times 0.61 \text{ mm}$) to better reveal the spatial details. The information of FOV is updated in Supplementary Table 1 along with the values of all the other experiments.

[References in Response]

- [R1] Hirvonen, L. M. *et al.* Lightsheet fluorescence lifetime imaging microscopy with wide - field time - correlated single photon counting. *Journal of Biophotonics* **13**, e201960099 (2020).
- [R2] Stellinga, D. *et al.* Time-of-flight 3D imaging through multimode optical fibers. *Science* **374**, 1395-1399 (2021).

REVIEWER COMMENTS

Reviewer #1 (Remarks to the Author):

The authors have made significant improvements to the quality of the paper in my view. The introduction is more concise, the technical presentation is sound and easy to follow, the merits clear and the results impressive. I support the publication of the manuscript, once the authors have addressed the following few points.

1. In the abstract the authors are referring to limitations in ultrafast videography techniques based on streak cameras, however there are several other methods that can trace events on ultrafast timescales that do not suffer from the mentioned limitations (the authors mention some of these in the introduction). Although methods based streak cameras have perhaps demonstrated more potential and a wider spread than other methods this statement is not entirely correct in my view and needs to be corrected.

2. I thank the authors for their discussion on the complex relationship between spatial resolution, imaging speed and temporal resolution in both the revised manuscript (lines 260-269) and Fig R1. I agree with most of what the authors say, in particular high imaging speed is essential to avoid the problems you show in Fig. R1c and I believe the temporal resolution offered by SCARF outperforms all current methods. But even if the data needs to be collected with a high temporal resolution to be able to accurately track rapid events (difference between Fig. R1b and c), the data might still be oversampled in time and the sequence depth thereby overestimated. If we consider the case in Fig. R1b, my point is that if the information in two neighbouring frames cannot be spatially resolved from each other due to poor imaging conditions, one of these frames is unnecessary and carries no valuable/additional information. This is not a problem per se, but it gives an incorrect value on the practical sequence depth. For the example in Fig. R1, the sequence depth should thus be reduced compared to the ground truth. This "problem" is still not clear in the manuscript. If the authors have already addressed this reduction in sequence depth in their data analysis, please make the readers aware of it.

3. Supplementary movie 6 does not show the line absorption pattern, as is written in the manuscript (line 204).

Response to Reviewer

We sincerely appreciate the reviewer for the thorough review of our work and the insightful and constructive comments, which help us improve the quality of our manuscript. In the following, we provide point-by-point responses. The changes in the revised manuscript are highlighted in red.

[Comment 0]

The authors have made significant improvements to the quality of the paper in my view. The introduction is more concise, the technical presentation is sound and easy to follow, the merits clear and the results impressive. I support the publication of the manuscript, once the authors have addressed the following few points.

[Response 0]

We thank the reviewer for acknowledging the improvement of the manuscript. We sincerely appreciate the reviewer's support for the publication of this manuscript.

[Comment 1]

In the abstract the authors are referring to limitations in ultrafast videography techniques based on streak cameras, however there are several other methods that can trace events on ultrafast timescales that do not suffer from the mentioned limitations (the authors mention some of these in the introduction). Although methods based streak cameras have perhaps demonstrated more potential and a wider spread than other methods this statement is not entirely correct in my view and needs to be corrected.

[Response 1]

Following the reviewer's suggestion and to enhance the clarity of the manuscript, we have revised the related part in Abstract. In particular, we have explicitly stated that the deployed optoelectronic devices can degrade the image quality and/or limit the imaging speed. We have also added a sentence to summarize the limitations of the other methods mentioned in Introduction. In this way, the statement in the revised Abstract is consistent with Introduction. Please see Lines 18–20 in the revised manuscript.

[Comment 2]

I thank the authors for their discussion on the complex relationship between spatial resolution, imaging speed and temporal resolution in both the revised manuscript (lines 260-269) and Fig R1. I agree with most of what the authors say, in particular high imaging speed is essential to avoid the problems you show in Fig. R1c and I believe the temporal resolution offered by SCARF outperforms all current methods. But even if the data needs to be collected with a high temporal resolution to be able to accurately track rapid events (difference between Fig. R1b and c), the data might still be oversampled in time and the sequence depth thereby overestimated. If we consider the case in Fig. R1b, my point is that if the information in two neighbouring frames cannot be spatially resolved from each other due to poor imaging conditions, one of these frames is unnecessary and carries no valuable/additional information. This is not a problem per se, but it gives an incorrect value on the practical sequence depth. For the example in Fig. R1, the sequence depth should thus be reduced compared to the ground truth. This "problem" is still not clear in the manuscript. If the authors have already addressed this reduction in sequence depth in their data analysis, please make the readers aware of it.

[Response 2]

We cherish this discussion with the reviewer, and we sincerely thank the reviewer for his/her careful reading of our explanation. We certainly agree with the reviewer if the data is temporally oversampled, the sequence depth can be overestimated. As the reviewer pointed out in his/her comment, this scenario can occur under poor imaging conditions. In our opinion, a key parameter to determine the imaging condition is the signal-to-noise ratio (SNR). At a given spatial resolution, a low SNR makes it difficult to distinguish the difference between two neighboring frames. Another key condition is the suitable speed of the imaging system for the duration of the transient event. Without satisfying this condition, even under perfect imaging conditions, a large number of reconstructed frames could be identical. This redundancy could arbitrarily inflate the sequence depth, which does not reflect the number of needed frames to effectively retrieve the dynamics in the scene.

We have taken into account these factors in this work. First, SCARF's probe pulse had a pulse energy of up to 1.6 mJ, which needed to be heavily attenuated to avoid saturation to the SCARF system during data acquisition in all experiments conducted in this work. This feature thus allowed obtaining a high SNR in the compressed snapshot by setting an appropriate optical attenuation.

The SCARF system also does not have a complicated light path, which ensures capturing a sharp compressed snapshot. Both contributors facilitated the reconstruction of high-quality images in the movie.

Meanwhile, we leveraged prior knowledge of the targeted transient events to optimize experimental conditions. Via literature review and theoretical calculation (e.g., Supplementary Note 6), we estimated the durations of the targeted transient events. The generation of these events and the SCARF measurement were synchronized so that we could always capture them. Then, we searched for an optimized combination of the imaging speed and the sequence depth with two criteria: (1) the evolution of the dynamic scene must be clearly resolved and (2) there must be discernable differences (in terms of movement and/or intensity changes) in neighboring frames recording the event's evolution. This search was executed by tuning the parameters of the chirped probe pulse. In each experiment, we tested several options and selected the one that produced the best quality movie.

Overall, using the approaches described above, we have ensured that each frame in the movie contains valuable and additional information, thus avoiding the temporal oversampling and arbitrary inflation in sequence depth in SCARF. Following the reviewer's suggestion and to make the readers aware of our effort, we have added this discussion to Lines 273–284 in the revised manuscript.

[Comment 3]

Supplementary movie 6 does not show the line absorption pattern, as is written in the manuscript (line 204).

[Response 3]

After carefully examining Supplementary Movie 6 in the submitted package, we found that the movie does show the line absorption pattern. Please see the following screenshots as proof. We speculate that maybe the reviewer was referring to the fact that the movie did not contain the 3D plot shown in Fig. 3e. Thus, we have updated Supplementary Movie 6 in the revised manuscript.

Step 1

Step 2

manuscripttrackingssystem nature communications

tracking system home submission guidelines reviewer inst

Detailed Status Information

Manuscript #	NCOMMS-23-08410
Current Revision #	1
Other Version	NCOMMS-23-08410
Submission Date	27th October 23
Current Stage	Awaiting resubmission
Title	Swept coded aperture real-time femtophotography
Manuscript Type	Article
Collection	Not applicable
Corresponding Author	Professor Jinyang Liang (jinyang.liang@inrs.ca) (Institut national de la recherche scientifique - Université du Québec)
Contributing Authors	Dr Jingdan Liu, Mr Yingming Lai, Dr Heide Ibrahim, Ms Katherine Légaré, Mr Xianglei Liu, Dr Philippe Lassonde, Professor Michel Hehn, Professor Stéphane Mangin, Dr Zhengyan Li, Gregory Malinowski, Professor François Légaré, Dr Miguel Marquez
Authorship	Yes
Abstract	Single-shot real-time femtophotography is indispensable to imaging ultrafast dynamics during their times of occurrence. Despite their advantages over conventional multiple-shot approaches, existing techniques confront degraded data quality by photoelectron imaging. They also face limitations in the acquirable information due to the space-time coupling in the sensing model. Here, we overcome these challenges by developing swept coded aperture real-time femtophotography (SCARF). By synergizing ultrafast laser sweeping, compressed sensing, and pulse shaping, SCARF enables all-optical ultrafast sweeping of a static coded aperture during the recording of an ultrafast event, which brings full-sequence encoding of up to 156.3 THz to every pixel on an ordinary CCD camera. We demonstrate SCARF's single-shot ultrafast imaging ability at tunable frame rates and spatial scales in both reflection and transmission modes. Using SCARF, we image ultrafast absorption in a semiconductor and ultrafast demagnetization of a metal alloy. With outstanding technical specifications and practical merits, SCARF is envisaged to make immediate contributions to ultrafast science.

We verify that the Manuscript # is correct and click on it.

Applicable Funding Source

Canada Foundation for Innovation (Fondation canadienne pour l'Innovation) - 37146 [Liang]
Fonds de Recherche en Nature et Technologies (Quebec Fund for Research in Nature and Technology) - 2019-NC-252960, 203345 - Centre d'optique, photonique et lasers [Liang]
Fonds de Recherche en Santé (Fonds de la recherche en santé du Québec) - 267405, 280229 [Liang]
New Frontier Research Fund (NFRF-2020-00267) [Liang]

Previous Interactions

None of the above

Transparent peer review

I have submitted the first version of my article, or transferred my article, to Nature Communications on or after 1st November 2022.

Policy and reporting checklists

Yes

Research Data Deposition

Yes

Figshare private link

<https://figshare.com/s/21e90a85eafa2a862c26>

Figshare DOI

<https://doi.org/10.6084/m9.figshare.24407194>

Manuscript Items

1. Author Cover Letter PDF (243KB)
2. "Response to Referees Letter" PDF (234KB)
3. Article File PDF (1895KB)
4. Video - Supplementary Movie 1 Video (14783KB)
5. Video - Supplementary Movie 2 Video (2809KB)
6. Video - Supplementary Movie 3 Video (3240KB)
7. Video - Supplementary Movie 4 Video (2593KB)
8. Video - Supplementary Movie 5 Video (4946KB)
9. Video - Supplementary Movie 6 Video (5914KB)
10. Video - Supplementary Movie 7 Video (772)
11. Reviewer Zip File "Zip of files for Reviewer"
12. Editorial Policy Checklist PDF (1390KB)

More Manuscript Info and Tools

Send Manuscript Correspondence
Decision Summary
Check Status

We click Supplementary Movie 6.

Step 3

In the opened Supplementary Movie 6, we see the generation of the line absorption pattern.

57.6 fs

Normalized intensity

REVIEWERS' COMMENTS

Reviewer #1 (Remarks to the Author):

The authors have now addressed all my questions in a satisfying way and I support the publication of the manuscript.

Response to Reviewer

[Comment]

The authors have now addressed all my questions in a satisfying way and I support the publication of the manuscript.

[Response]

We thank the reviewer for acknowledging that we have addressed all the questions in a satisfying way and for supporting the publication of this manuscript.